# How Satiating Are the ‘Satiety’ Peptides: A Problem of Pharmacology versus Physiology in the Development of Novel Foods for Regulation of Food Intake

**DOI:** 10.3390/nu11071517

**Published:** 2019-07-04

**Authors:** Jia Jiet Lim, Sally D. Poppitt

**Affiliations:** 1Human Nutrition Unit, School of Biological Sciences, University of Auckland, Auckland 1024, New Zealand; 2Riddet Institute, Palmerston North 4442, New Zealand; 3Human Nutrition Unit, Department of Medicine, School of Biological Sciences, University of Auckland, Auckland 1024, New Zealand

**Keywords:** appetite, satiety, cholecystokinin, glucagon-like peptide-1, peptide YY, dietary studies, infusion studies

## Abstract

Developing novel foods to suppress energy intake and promote negative energy balance and weight loss has been a long-term but commonly unsuccessful challenge. Targeting regulation of appetite is of interest to public health researchers and industry in the quest to develop ‘functional’ foods, but poor understanding of the underpinning mechanisms regulating food intake has hampered progress. The gastrointestinal (GI) or ‘satiety’ peptides including cholecystokinin (CCK), glucagon-like peptide 1 (GLP-1) and peptide YY (PYY) secreted following a meal, have long been purported as predictive biomarkers of appetite response, including food intake. Whilst peptide infusion drives a clear change in hunger/fullness and eating behaviour, inducing GI-peptide secretion through diet may not, possibly due to modest effects of single meals on peptide levels. We conducted a review of 70 dietary preload (DIET) and peptide infusion (INFUSION) studies in lean healthy adults that reported outcomes of CCK, GLP-1 and PYY. DIET studies were acute preload interventions. INFUSION studies showed that minimum increase required to suppress ad libitum energy intake for CCK, GLP-1 and PYY was 3.6-, 4.0- and 3.1-fold, respectively, achieved through DIET in only 29%, 0% and 8% of interventions. Whether circulating ‘thresholds’ of peptide concentration likely required for behavioural change can be achieved through diet is questionable. As yet, no individual or group of peptides can be measured in blood to reliably predict feelings of hunger and food intake. Developing foods that successfully target enhanced secretion of GI-origin ‘satiety’ peptides for weight loss remains a significant challenge.

## 1. Introduction

Developing novel foods that can enhance satiety and reduce overconsumption in overweight individuals is an important target in the quest for weight loss. It has long been proposed that the satiating effect of a food can be optimised by modifying components including the macronutrient composition and physicochemical structural properties [1], and clearly novel food products where efficacy can be demonstrated are likely to be of both public health and commercial value in the current environment. However, a better understanding of the causal mechanisms that regulate food intake is required to progress this area. Despite a significant and growing literature, fundamental questions regarding the physiological regulation of food intake remain unanswered. 

The focus of this review is the gastrointestinal (GI)-derived, or commonly termed ‘satiety’, peptides, which include cholecystokinin (CCK), glucagon-like peptide-1 (GLP-1) and peptide YY (PYY). They are secreted from both the proximal and distal intestine in response to the arrival of nutrients into the intestinal tract [2,3,4,5,6,7]. Certainly, these peptides play a role in digestion and absorption of foods within the GI tract [8,9,10]. These peptides have long been purported to be predictive biomarkers which can provide a snapshot of subjective feelings of appetite and future food intake. Although the increase in circulating GI peptide concentrations is observed to occur concurrent with suppression of appetite following ingestion of a meal, the increased circulating GI peptides in turn drive appetite-related responses has yet to be satisfactory demonstrated in dietary intervention trials (Figure 1). Another major challenge faced by dietary preload studies looking to support this hypothesis is the need to disentangle the direct effect of GI peptides on hunger suppression from other potential anorectic stimulants associated with food intake.

Robust evidence from dietary intervention trials is required to substantiate any appetite-related claims to be made on food products. A standardised international methodology for assessing acute satiety effects of food ingredients, products and meals was described by the International Life Sciences Institute (ILSI) working group led by Blundell, et al. [11] as the dietary preload study. This study design requires consumption of a fixed meal as dietary preload, with postprandial response assessed through measurement of subjective feeling of appetite, such as hunger or fullness, using visual analogue scales (VAS), followed by measurement of ad libitum food intake at an outcome meal at a fixed time interval post-preload. The European Food Safety Authority (EFSA) recognises VAS-assessed change in appetite ratings and energy intake as an appropriate outcome measure of mechanisms in support of weight loss (EFSA 2012). Therefore, we were interested to review the current dietary literature investigating trials that have adhered to this EFSA standard to determine whether optimising a food product to target the ‘satiety’ peptides is a reasonable strategy for development of novel foods.

In contrast to the dietary studies, intravenous infusion of GI peptides in the fasted state leads to a significant and rapid increase in circulating GI peptide concentration, and there is a clear suppression of food intake. These studies, presented in detail later in this review and commonly known as peptide infusion studies, have collectively shown that GI peptide receptors can be potential therapeutic targets for obesity, with peptide signalling pathways also being elucidated [12,13]. Anti-obesity drugs targeting peptide receptors show some efficacy in clinical studies [14]. GLP-1 receptor agonists are well known for treatment of type 2 diabetes [15] and also show some promise for weight control [16,17]. Nevertheless, these are pharmacological mechanisms. Essentially, appetite suppression is generally more pronounced in peptide infusion studies than in dietary preload studies, and most importantly, circulating GI peptides are characteristically higher following infusion than following a meal [18]. The question on physiological mechanism, whether postprandial increase in GI peptides causes the suppression of food intake or not, is not satisfactorily answered by peptide infusion studies.

Mars, Stafleu and de Graaf [18] previously reviewed the utility of measuring circulating GI peptides in appetite studies and was unable to conclude that circulating GI peptides are useful indicators of appetite. Their results also challenged the relevance of developing novel food products targeting GI peptides as a mechanism by which to suppress daily energy intake. Our current review builds on this earlier analysis [18] and aims to further explore the relationship between GI peptides and appetite outcomes, in addition to discussing the relevance of targeting mechanisms of GI peptides during the development of novel food products. The primary objective of this review was to assess whether there were differences in baseline peptide concentration, C_max_, and fold change from baseline between dietary preload (DIET) studies and peptide infusion (INFUSION) studies. The secondary objective was to explore whether there was an association between peptide concentration and appetite outcomes in both groups.

## 2. Materials and Methods

An online literature search was conducted by J.J.L. for relevant articles published on PubMed. Original studies were identified by using the following keywords: “energy intake” or “appetite” or “satiety” or “eating”, in combination with “cholecystokinin” or “glucagon-like peptide-1” or “peptide YY” or their abbreviations (CCK, GLP-1 and PYY respectively). Relevant articles were initially selected based on the title and abstract. The articles selected were required to be original studies conducted as either dietary preload (DIET) studies or peptide infusion (INFUSION) studies, written in the English language. Initial searches were carried out for articles published between year 2003 and 2017. However, due to the small number of published of peptide infusion studies, searches for peptide infusion studies were extended to 1993. The references of selected articles were also examined to identify any further relevant articles.

Studies were selected based on the following criteria: (i) Dietary preload study conducted using a standardised preload design, as described by Blundell et al. [11] where a fixed preload was consumed, and an outcome meal provided after a fixed time interval from which the participant could eat freely (*ad libitum*); or peptide infusion study conducted using intravenous peptide infusion as treatments and saline infusion as control, (ii) human clinical studies, (iii) lean cohort with mean BMI < 25.0 kg/m^2^, (iv) no clinically diagnosed disorders. The studies were included if baseline and peak (C_max_) concentration of CCK, GLP-1 or/and PYY were reported in text or could otherwise be extracted from Figures showing change of concentration across time. Some missing data were obtained from the authors. Studies were excluded if interventions were targeted towards trained athletes, or centred on a specific age range, such as children, young adults or elderly, as these groups did not represent a typical lean and healthy adults. Furthermore, studies which allowed participants to request the outcome meal spontaneously were excluded due to the difficulty in characterising mean C_max_ of GI peptides prior to consumption of the outcome meal. Studies were also excluded if they involved additional intervention, such as exercise. Additionally, any preload meals containing less than 1 MJ were excluded, as they represented a small meal.

Data collected include the duration between preload meal and outcome meal, energy and macronutrient composition of preload meal, baseline peptide concentrations, peak peptide concentrations, and *ad libitum* energy intake at the outcome meal. Where peptide concentrations were only available in the form of graphs, this was manually measured using the PDF-measurement tool (Adobe Acrobat Pro DC, Burlington, NJ, USA). This tool enables the measurement of the distance between two points on the graph to the lower detection limit of 0.01 cm. Fold change in relative to baseline was calculated by dividing the C_max_ by the baseline concentration. Preload meal was grouped into either solid (e.g., sandwiches and composite meal) or non-solid (e.g., liquid drinks, soup, pudding and custard). Data were analysed using IBM Statistical Package for the Social Sciences (SPSS) software (version 25; IBM Corp., Armonk, NY, USA). Weighted means and medians were calculated for each peptide (CCK, GLP-1 and PYY), and grouped into dietary preload studies (DIET) or peptide infusion studies (INFUSION). Weight was applied to each intervention based on its sample size (*n*); therefore, a greater emphasis was given to an intervention with greater sample size. Statistical outliers were not removed from the analysis unless otherwise specified. A dot plot overlaid on a box plot was presented to demonstrate the data distribution; each dot was representative of the mean of an intervention. Data were reported as ‘mean (95% Confidence Interval)’ unless otherwise specified. *N* represents the total sample size while *K* represents the number of interventions.

## 3. Results

### 3.1. Search Results

Based on the search methods described, 52 DIET articles were retrieved in full text. Of 52 articles, 9 articles were excluded due to insufficient data. Then, two articles were further excluded due to duplicate data [19] and unreliable data [20]. As a result, a total of 41 DIET articles were included in the analysis. A total of 18 INFUSION articles were included in the analysis (Figure 2). 

### 3.2. Study Characteristics

DIET comprised of 108 dietary interventions published in 41 articles (Table 1). All studies were randomised cross-over trials (≥ 2 interventions in a single article). Fifty-seven interventions (53%) had mixed gender, whereas 34 interventions (31%) enrolled male participants only and 17 interventions (16%) enrolled female participants only. The median sample size of the intervention was 17 (range: 4–40). The median duration of DIET was 180 min (range: 30–480 min). The preload meals varied from study to study, from solid to non-solid format. The median energy supplied by the preload meal was 1.8 MJ (range: 1–12.5 MJ). The energy content of preload meals supplied by 4 interventions [21,22] were adjusted based on the participants energy requirement; whereas in other studies, preload meals supplied fixed energy to all participants within an intervention. All interventions required the preload meal to be consumed in its entirety. Each dietary intervention assessed at least one GI peptide, whilst several interventions assessed multiple peptides. For DIET, in the 41 published articles, a total of 39 interventions assessed CCK, 66 interventions assessed GLP-1 and 50 interventions assessed PYY. 

INFUSION consisted of 34 intravenous infusion interventions published in 18 articles (Table 2). All studies were randomised controlled trials. Each study consisted of 1 to 6 interventions, excluding the control (placebo saline infusion). Eight interventions (24%) had a combination of genders, whereas 25 (74%) interventions enrolled male participants only and 1 (3%) intervention enrolled female participants only. Notably, male were over-represented in INFUSION. The median sample size of INFUSION was 10 (range: 6–24). There were a total of 8 interventions which infused CCK, 11 interventions which infused GLP-1 and 15 interventions infused PYY.

The methodology of INFUSION was less standardised than DIET. Variability included infusion duration and presence of oral preload. Eighteen (53%) interventions infused from the fasting state until completion of the outcome meal, whereas 13 (38%) interventions stopped the infusion prior to the meal. The outcome meal was not part of the procedure in 3 (9%) interventions. The median infusion duration was 90 min (range: 10–240 min). In 63% of the CCK infusions, and in 46% of the GLP-1 infusions, an oral preload was given prior to or during the infusion. None of the PYY infusions had an oral preload. Gastric cues were believed to play a role in lowering the threshold concentration at which CCK exhibits its physiological effects on satiety. Banana and water were the most common dietary preload in CCK infusion studies, because these preloads were not found to induce the secretion of GI peptides significantly [61,62]. All peptide infusion studies had placebo saline infusion as a control, but the results are not presented in this review. Generally, DIET had a larger sample size, were of longer duration and followed a more standardised protocol than INFUSION.

**Table 2 nutrients-11-01517-t002:** Peptide infusion (INFUSION) studies in lean men and women.

Reference	Gender	*N*	Oral Preload (mins)	Infusion Duration (mins)	Peptides	Dosage	Fold Change	Time of *ad libitum* Meal (mins)	Appetite Outcomes	
									Sensation	FI
Lieverse, et al. [63]	MF	9	—	*t* = 0–135	CCK-33	0.2 pmol/kg ideal BW/min	5.6	*t* = 60	ND	−12%

Ballinger et al. [62]	MF	6	200 mL water (*t* = 20)	*t* = 0–40	CCK-8	0.54 pmol/kg/min	16.2	*t* = 75	—	−21% *

Lieverse et al. [61]	F	10	552 kJ banana shake (*t* = 60)	*t* = 0–165	CCK-33	0.2 pmol/kg/min	4.4	*t* = 75	ND	−18% *

Gutzwiller, et al. [64]	M	16	644 kJ banana shake (*t* = −20)	*t* = −5–5	CCK-8	67.5 pmol/min	4.9	*t* = 0 – 60	HGR▼, FUL▲	−7%

MacIntosh, et al. [65]	MF	12	744 kJ banana shake (*t* = 90)	*t* = 100–125	CCK-8	0.9 pmol/kg/min	18.7	*t* = 140	ND	−1%
	MF	12	744 kJ banana shake (*t* = 90)	*t* = 100–125	CCK-8	2.7 pmol/kg/min	37.8	*t* = 140	ND	−29% *

Gutzwiller, et al. [66]	M	24	—	*t* = −60–60	CCK-33	0.2 pmol/kg/min	3.6	*t* = 0	ND	−11% *
	M	24	—	*t* = −60–60	GLP-1_active_	0.9 pmol/kg/min	4.0	*t* = 0	ND	−9% *

Brennan, et al. [67]	M	24	—	*t* = 0–150	CCK-8	1.8 pmol/kg/min	3.8	*t* = 120	FUL▲	−23% *
	M	24	—	*t* = 0–150	GLP-1_active_	0.9 pmol/kg/min	3.8	*t* = 120	ND	+1%

Flint, et al. [68]	M	19	fixed meal (*t* = 0)	*t* = 0–240, 270–300	GLP-1_total_	0.83 pmol/kg/min	8.4	*t* = 120	STT▲, HGR▼, FUL▲	−12% *

Gutzwiller, et al. [69]	M	16	—	*t* = 0–80	GLP-1_active_	0.375 pmol/kg/min	4.6	*t* = 60	HGR ^a^	−7%
	M	16	—	*t* = 0–80	GLP-1_active_	0.75 pmol/kg/min	6.3	*t* = 60	HGR ^a^	−11% *
	M	16	—	*t* = 0–80	GLP-1_active_	1.5 pmol/kg/min	14.9	*t* = 60	HGR▼ ^b^	−32% *

Long, et al. [70]	M	10	400 mL water (*t* = 20)	*t*= 0–60	GLP-1_total_	1.2 pmol/kg min	9.7	*t* = 40	ND	−7%

Nagell, et al. [71]	NF	8	300 mL beef tea (*t* = 15)	*t* = 0–60	GLP-1_total_	0.5 pmol/kg/min	4.4	—	HGR▼	—

Neary, et al. [72]	MF	10	—	*t* = 0–120	GLP-1_total_	0.4 pmol/kg/min	2.9	*t* = 90	—	−5%
	MF	10	—	*t* = 0–120	PYY	0.4 pmol/kg/min	6.7	*t* = 90	—	−15%

Little, et al. [73]	M	10	100 g minced beef tea (*t* = 15)	*t* = −30–120	GLP-1_total_	0.3 pmol/kg/min	2.5	—	ND	—
	M	10	100 g minced beef tea (*t* = 15)	*t* = −30–120	GLP-1_total_	0.9 pmol/kg/min	4.3	—	ND	—

Batterham, et al. [74]	MF	12	—	*t* = 0–90	PYY	0.8 pmol/kg/min	5.2	*t* = 210	HGR▼	−33% *

Degen, et al. [75]	M	16	—	*t* = −60–30	PYY	0.2 pmol/kg/min	2.1	*t* = 0	HGR ^a^	−7%
	M	16	—	*t* = −60–30	PYY	0.4 pmol/kg/min	3.1	*t* = 0	HGR ^a^	−11% *
	M	16	—	*t* = −60–30	PYY	0.6 pmol/kg/min	5.1	*t* = 0	HGR▼ ^b^	−32% *

le Roux et al. [35]	M	6	—	*t* = 0–90	PYY	0.2 pmol/kg/min	2.3	*t* = 210	FUL ^a^	+2%
	M	6	—	*t* = 0–90	PYY	0.4 pmol/kg/min	3.6	*t* = 210	FUL ^a^	−6%
	M	6	—	*t* = 0–90	PYY	0.5 pmol/kg/min	4.3	*t* = 210	FUL▲ ^b^	−12%
	M	6	—	*t* = 0–90	PYY	0.6 pmol/kg/min	4.8	*t* = 210	FUL▲ ^b^	−16%
	M	6	—	*t* = 0–90	PYY	0.7 pmol/kg/min	5.5	*t* = 210	FUL▲ ^b^	−22% *
	M	6	—	*t* = 0–90	PYY	0.8 pmol/kg/min	6.8	*t* = 210	FUL▲ ^b^	−23% *

Batterham, et al. [76]	M	8	—	*t* = 0–90	PYY	0.8 pmol/kg/min	2.3	*t* = 120	PCF▼	−25% *

le Roux, et al. [77]	M	6	—	*t* = 0–90	PYY	1.0 pmol/kg/min	11.2	*t* = 210	STT▲	−18% *
	M	6	—	*t* = 0–90	PYY	1.0 pmol/kg/min	7.2	*t* = 210	STT▲	−21% *
	M	6	—	*t* = 0–90	PYY	1.0 pmol/kg/min	6.9	*t* = 210	STT▲	−20% *


In some studies, a small oral preload (<1 MJ) was given to participants. The time reported refers to the time point at which oral preload was given, no information about the time given to consume the preload completely was found in the original articles. The time reported for the infusion duration represents the starting and ending time point of the peptide infusion, the study may continue after the infusion until ad libitum meal. The time reported for *ad libitum* meal represents the time point at which the meal was given to participants; no information about the time given to consume the meal was found in original articles for most studies. Appetite outcomes that are statistically different from each other are indicated by different superscript letters, i.e., a and b. Only subjective appetite outcomes that are statistically different when expressed in terms of Area under the Curve (AUC) are reported, unless otherwise specified. Effects on food intake was compared with placebo control and reported as increase/decrease in percentage energy intake. Abbreviations and symbols: M = Male, F = Female, MF = Mixed gender, FI = Food intake, ND = No significant difference, FUL = Fullness, STT = Satiety, HGR = Hunger, PCF = Prospective consumption of food, ▲ = Significant increase when compared to placebo control, ▼ = Significant decrease when compared to placebo control, * = Food intake significantly different from placebo control, — = Oral preload, *ad libitum* meal, or subjective appetite assessment was not included in the study.

### 3.3. CCK

CCK is a group of molecules consisting of various molecular forms, named after their peptide chain length. CCK-8 and CCK-33 have 8 and 33 amino acids respectively. They are 2 of the most common molecular forms measured in appetite studies, especially in INFUSION. In this review, all forms of CCK were grouped into one analysis, as the form of CCK being assessed was not made clear in most studies. The range of CCK baseline concentration (Figure 3a) was similar between DIET (0.17–8.30 pM) and INFUSION (0.45–7.90 pM), as expected. However, the weighted means had a small but significant difference between DIET (1.72 pM, 95% CI: 1.59–1.84 pM) and INFUSION (2.38 pM, 95% CI: 1.95–2.81 pM) (*p* < 0.01). Despite the wide range, it is noticeable from Figure 3a that most of the data was distributed below 4.00 pM for both DIET and INFUSION. The range of CCK **C_max_** (Figure 3b) was much narrower for DIET (0.90–9.40 pM) than INFUSION (2.55–45.40 pM). The weighted mean was also significantly lower for DIET (3.34 pM, 95% CI: 3.17–3.51 pM) than INFUSION (17.77 pM, 95% CI: 15.07–20. 46 pM) (*p* < 0.01). Six of 8 INFUSION interventions resulted in CCK **C_max_** levels unachievable by DIET interventions. The fold change of CCK (Figure 3c) ranged from 0.51-fold to 7.85-fold for DIET, much narrower than INFUSION, which ranged from 3.60-fold to 37.80-fold. The weighted mean was also significantly lower for DIET (2.98-fold, 95% CI: 2.83–3.13-fold) than INFUSION (10.96-fold, 95% CI: 8.68–3.23-fold) (*p* < 0.01). Attention was given to two DIET articles that reported a 7.85- and 7.06-fold increase in CCK [34,39]. However, the form of CCK analysed was CCK-8, had low baseline concentration (range: 0.17–0.34 pM) and a relatively low absolute **C_max_** (range: 0.90–1.57 pM). Surprisingly, these two studies did not show greater fold change in CCK concentrations when compared to their active-controlled interventions (7.85-fold vs. 3.94-fold, 7.06-fold vs. 5.29-fold) significantly decreased subjective appetite sensation and food intake.

### 3.4. GLP-1

There are two forms of GLP-1, active and total. GLP-1 is secreted in active form into circulation, where it is highly susceptible to deactivation by dipeptidyl peptidase IV (DPP-IV) enzyme. Active GLP-1 has a very short half-life of 1 min [78,79]. In this review, GLP-1 is grouped individually into active and total GLP-1 for the assessment of baseline and **C_max_**, and pooled together when assessing fold change. This method, which assumes that relative increase in active GLP-1 is equal to relative increase in total GLP-1, also allowed comparison with the prior review [18]. The range of total GLP-1 baseline concentration (Figure 4a) was much greater for DIET (range: 9.40–40.60 pM) than INFUSION (10.00–21.50 pM). This was unexpected. The weighted mean was also significantly higher for DIET (17.78 pM, 95% CI: 17.25–18.31 pM) than INFUSION (12.48 pM, 95% CI: 11.52–13.44 pM) (*p* < 0.01). A DIET study conducted by Dougkas and Ostman [20] reported that total GLP-1 baseline concentration ranged between 105.90 pM and 162.70 pM, at least 15 standard deviations greater than the weighted mean in this analysis. They were being removed from our analysis as data may be unreliable (see Figure 2). The range of total GLP-1 **C_max_** (Figure 4b) for DIET (19.40–98.60 pM) was similar to that of INFUSION (25.70–126.00 pM). However, the weighted mean was significantly lower for DIET (32.59 pM, 95% CI: 31.27–33.91 pM) than INFUSION (69.30 pM, 95% CI: 61.37–77.24 pM) (*p* < 0.01).

The range of the active GLP-1 baseline concentration (Figure 5a) for DIET was 3.60–13.00 pM, whereas the range for INFUSION was 0.60–7.00 pM. Similar to total GLP-1 baseline concentration, the weighted mean was significantly higher for DIET (4.77 pM, 95% CI: 4.68–4.86 pM) than INFUSION (1.43 pM, 95% CI: 0.99–1.87 pM). This was again unexpected. Notably, 22 of 29 DIET interventions assessing active GLP-1 were conducted by Westerterp and colleagues [21,22,41,42,43,44]; whereas 4 of 5 INFUSION interventions were conducted by Gutzwiller and colleagues [66,69]. Results reported by the same group of researchers had reasonably consistent baseline concentration. The range of active GLP-1 C_max_ for DIET was 6.10–25.00 pM, whereas the range for INFUSION was 2.40–35.00 pM (Figure 5b). The weighted means were not significantly different between DIET (8.73 pM, 95% CI: 8.49–8.97 pM) and INFUSION (8.67 pM, 95% CI: 6.46–10.87 pM) (*p* = 0.96). Although ranges and means were similar, the higher baseline concentration reported in DIET should not be overlooked during interpretation.

The range of GLP-1 fold change (total and active combined) (Figure 6) was much narrower for DIET (1.09–3.63-fold) than INFUSION (2.50–14.90-fold). The weighted mean was also significantly lower for DIET (1.85-fold, 95% CI: 1.82–1.89-fold) than INFUSION (6.29 pM, 95% CI: 5.72–6.86 pM) (*p* < 0.01). Sanggaard and colleagues [29] reported the highest GLP-1 fold change among DIET (3.39-fold and 3.69-fold), however, the preload energy content was 3.7 MJ and 4.1 MJ, respectively, approximately 2× greater than median energy intake (1.8 MJ) in this review. Also notable was the DIET study conducted by Juvonen and colleagues [47], which showed greater fold change in GLP-1 (3.57-fold) following consumption of whey protein beverage than 2 casein protein beverages (1.92-fold and 1.40-fold). Despite greater fold change, there was no significant difference in subjective appetite sensations.

### 3.5. PYY

Unexpectedly, the range of PYY baseline concentration (Figure 7) for DIET (2.20–97.00 pM) was greater than INFUSION (8.30–28.70 pM). The weighted mean was also significantly higher for DIET (32.28 pM, 95% CI: 30.27–34.29 pM) than INFUSION (76.41 pM, 95% CI: 70.13–82.69 pM) (*p* < 0.01). The range of PYY **C_max_** was greater for DIET (3.30–170.00 pM) than INFUSION (24.90–147.90 pM). The weighted mean was also significantly lower for DIET (47.72 pM, 95% CI: 44.86–50.58 pM) than INFUSION (76.41 pM, 95% CI: 70.13–82.69 pM) (*p* < 0.01). Two DIET studies [46,54], which reported very high **C_max_** (range: 100.00–170.00 pM) also reported very high baseline concentrations (range: 77.00–97.00 pM). The range of PYY fold change was lower for DIET (0.78–4.17-fold) than INFUSION (2.13–11.23-fold). The weighted mean was also significantly lower for DIET (1.82-fold, 95% CI: 1.77–1.88-fold) than INFUSION (4.78 pM, 95% CI: 4.40–5.16 pM) (*p* < 0.01). le Roux and colleagues [35] demonstrated that secretion of PYY increased with energy consumption. Another noteworthy study involved the consumption of 1030 mg of red pepper (containing 80,000 Scoville Heat Units of capsaicin) with a lunch meal, which led to a 4.11-fold increase in PYY secretion, compared to 2.17-fold for placebo [21]. However, no statistical difference in subjective appetite sensations was reported in this study.

## 4. Discussion

Despite a long history of appetite research, a number of fundamental questions on the physiological regulation of food intake remain unanswered. One key question is whether peptides secreted from the GI tract following consumption of a meal are reliable biomarkers of hunger, fullness and other appetite-related responses, or more importantly subsequent food intake. Our review of the literature concludes that causal links between food consumption and the appearance of GI-secreted peptides in peripheral circulation at concentrations likely to drive changes in appetite response and eating behaviour remain to be demonstrated. As yet, no single peptide or group of peptides can be measured in blood to predict how hungry you feel or what and how much you are likely to eat at your next meal. 

This review presents data on baseline and peak concentration in addition to fold change in 3 key circulating peptides, i.e., CCK, GLP-1 and PYY, in response to both peptide infusion and consumption of a meal in healthy adults. The great variability in baseline concentration was unexpected since we included only studies of lean and healthy adults, and as a result, it was not of value to review absolute peak concentration between studies. Whilst the range of peak concentration in the dietary studies provided valuable information regarding the physiological postprandial range, relative change between baseline and peak concentration (fold change) allowed comparison between studies. An important finding of this review was that the postprandial fold change of CCK, GLP-1 and PYY following food intake was consistently lower for all peptides when compared to that observed following exogenous peptide infusion. 

### 4.1. Exploring the Relationship between Circulating GI Peptides and Appetite Outcomes

Based on the existing literature, this current review found no clear evidence that the postprandial fold change in peripheral circulating GI peptide concentrations in response to an energy bolus of at least 1 MJ was linked to appetite sensation in lean and healthy adults. A linear relationship between fold change in GI peptides and ad libitum energy intake could not be established in dietary preload studies. Consequently, the diet-induced increase in GI peptides was not consistent with appetite outcomes. Based on the peptide infusion studies, the minimum fold change reported to decrease ad libitum energy intake for each of CCK, GLP-1 and PYY was 3.6, 4.0 and 3.1-fold, respectively. In other words, infusing GI peptides to meet these ‘thresholds’ may result in significant suppression of ad libitum energy intake when compared to saline infusion. In contrast, no evidence of significant suppression of ad libitu energy intake was found when the infusion was insufficient to meet these ‘thresholds’. Only 29% of reported CCK concentrations in dietary studies met this ‘threshold’ fold change, whilst this was even lower at 0% for GLP-1 and 8% for PYY. This observation confirms that it is very challenging to elevate postprandial circulating GI peptides to meet the ‘threshold’ identified from peptide infusion studies using a low-energy preload. The energy content of preloads in this review ranged predominantly between 1 to 4.7 MJ (outliers 8.4, 12.5 MJ), with mean preload of 2.3 MJ. However, the ‘threshold’ could potentially be lowered as had been well demonstrated by Lieverse and colleagues. They showed that gastric distention can lower the CCK threshold to induce a satiety response [61,63]. Hence, the ‘actual threshold’ of GI peptides could potentially be lower than those that we propose in this review, based on the following two arguments.

Firstly, the ‘threshold’ suggested in this review was based on the ‘single-peptide approach’, which investigated the fold change of a single peptide as an independent factor for ad libitum energy intake. Most peptide infusion studies were carried out by infusing a single peptide at a time. While this approach is useful for excluding confounders, the interaction between 2 or more GI peptides should not be overlooked. Some peptide infusion studies have infused 2 peptides simultaneously [66,67,72,80]. There was some evidence of peptides acting additively or synergistically [72,80,81], providing a strong rationale to measure 3 peptides in a single dietary study. Unfortunately, the data to date is too scarce to draw any conclusions.

Secondly, intravenous GI peptide infusion into systemic circulation may bypass a potentially significant paracrine or neurocrine stimulation pathway, which happens around the site of secretion and vagal afferent nerves, before hepatic clearance takes place [82]. Based on the data from peptide infusion studies, these GI peptides are potent endocrine hormones; however, their full potency may lie beyond endocrine activities. There is evidence that CCK acts via paracrine pathway before being diluted in the systemic circulation, as shown in an early study where hepatic-portal infusion of CCK was less efficient than intraperitoneal infusion of CCK in suppressing energy intake [83]. Also, it has been proposed that GLP-1 triggers a satiety response by activating GLP-1 receptors near the site of secretion before entering the systemic circulation, although the endocrine effect of GLP-1 was also acknowledged [84]. In contrast to CCK and GLP-1, much research on PYY has focused on Y_2_ receptors in the central nervous system [12].

These arguments raise the possibility that the ‘threshold’ observed in in this review may be too high. Irrespective, there is insufficient evidence to link the circulating GI peptides to appetite response at a lower circulating concentration.

### 4.2. Implications for the Development of Satiety-Enhancing Novel Food Products

Developing satiety-enhancing novel food products by targeting the mechanisms of GI peptide secretion requires delivery of macronutrients into the intestine. The ability of available carbohydrate, protein and fat to induce secretion of GI peptides has been characterised in studies which infused single nutrients into different regions of the intestinal tract [2,3,4,5,6,7]. However, drawing implications on the dose-dependent secretion of GI peptides based on these findings has been unsuccessful. For example, despite the satiating effect of dietary protein, which is well supported through dietary studies [48,85], no linear relationship was observed between protein content and GI peptide response in our current review. The considerable variability in macronutrient composition, energy density, food format (solid or liquid), texture and other preload variables may in part explain this. It is also noteworthy that GI peptide secretion can be modified by unavailable carbohydrate, i.e., dietary fibre [36,86,87,88]. Dietary fibre in turn modifies the delivery of available carbohydrate, protein and fat to the nutrient-sensing cells, and is well known for its satiating effect [89,90]. Strategies for delivering undigested nutrients into the distal intestine in order to stimulate secretion of GLP-1 and PYY appear to be extremely challenging to deploy [2]. Unlike postprandial changes in peptides such as insulin, postprandial changes in the GI-derived ‘satiety’ peptides are more subtle and difficult to predict based on existing studies. 

### 4.3. Limitations and Recommendations for Future Studies

One of the limitations of our findings is that this is not a systematic review. Although reporting PubMed as the sole database may have resulted in fewer studies, the main outcome of the analysis is unambiguous and unlikely to change significantly with the addition of further postprandial studies.

Secondly, total concentration of circulating GI peptides over the postprandial period, reported as AUC, would be a more preferable measure when comparing differential peptide responses within studies. However, since studies differed in duration from 30 to 480 min, AUC was not an appropriate measure to use; hence, baseline and peak concentration were analysed. Future reviews could investigate the effect of peptide concentration prior to the ad libitum outcome meal on the energy intake. 

Thirdly, although our current review showed poor associations between circulating concentration of single GI peptides and appetite outcomes, it is possible that measuring multiple peptides CCK, GLP-1 and PYY concurrently may be more informative and improve the understanding of additive or synergistic mechanisms of peptides in the regulation of appetite. This could potentially lead to profiling an appetite-related biomarkers ‘fingerprint’. Studies which investigate longer-term circulating peptide concentrations over periods of 24 h [91] or several days may also be more informative than single meal outcomes.

The potential role of phenotypic or individual variability on GI peptides profile and satiety responses has been presented in previous reviews [14,82,92]. Hence, finally, since our current review targeted only a healthy, lean adult population, our findings are not applicable to obese populations which may have a unique GI peptides profile, although notable this is not without conflicting results [53]. Clamp and colleagues [57] recently demonstrated that obese individuals who habitually consume a high-fat diet had a lower postprandial PYY response than lean individuals, yet similar *ad libitum* energy intake between the two groups. Giezenaar and colleagues also recently demonstrated that young men had greater postprandial GLP-1 and PYY, accompanied by a greater suppression in *ab lib* energy intake than young women [93]; but lower postprandial CCK than old men, accompanied by a lesser suppression in *ad libitum* energy intake [5]. Collectively, the variable appetite response to an identical preload meal is an emerging concept known as the satiety phenotype [94].

## 5. Conclusions

Long-term weight loss remains an unattainable goal for many individuals, despite the considerable resource invested in the development of dietary products and strategies for treatment of obesity. Improved understanding of the mechanisms underpinning appetite response and eating behaviour is likely required in order to develop novel modified or ‘functional’ foods that can successfully target overconsumption and aid weight loss. Whilst optimising food products to target circulating concentrations of GI ‘satiety’ peptides with intent to suppress hunger and desire to eat and promote fullness appears to be a reasonable strategy, our review of dietary studies shows that this may be a difficult target to achieve. These peptides are rapidly secreted postprandially, with numerous physiological roles including that of nutrient digestion and absorption, but whether they are causal in suppression of appetite and eating behaviour at the low circulating levels attained following a meal is far less clear. 

## Figures and Tables

**Figure 1 nutrients-11-01517-f001:**
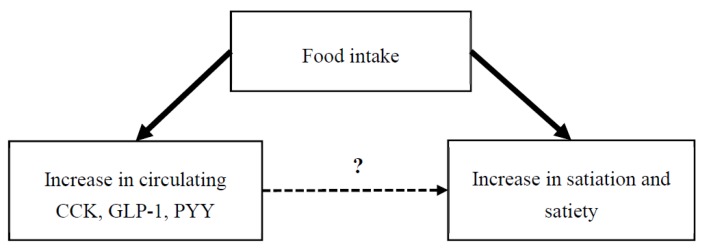
Food intake induces an increase in circulating gastrointestinal peptides CCK, GLP-1 and PYY, as well as parallel increase in satiation and satiety. However, whether these ‘satiety’ peptides in turn elicit a direct physiological effect on aspects of eating behaviour is less well-understood.

**Figure 2 nutrients-11-01517-f002:**
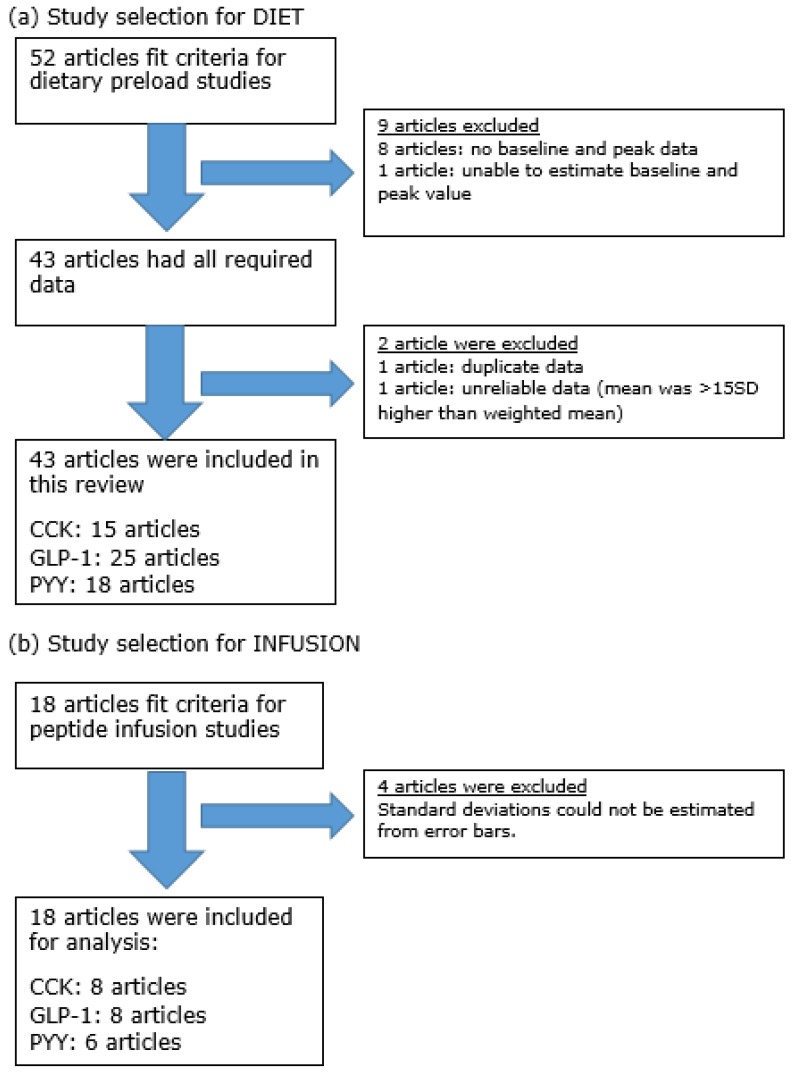
Flow diagram of article selection for (**a**) DIET and (**b**) INFUSION studies. Some articles reported >1 GI-peptide.

**Figure 3 nutrients-11-01517-f003:**
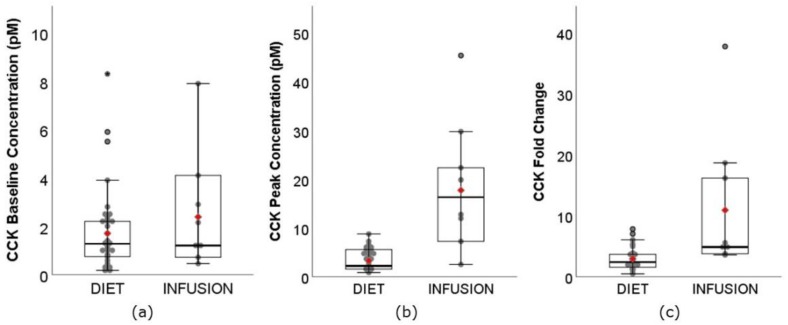
Boxplots showing the (**a**) baseline concentration, (**b**) **C_max_**, peak concentration, and (**c**) fold change of CCK between DIET (N = 620, K = 39) and INFUSION (N = 98, K = 8). The weighted means were significantly different between DIET and INFUSION for all (*p* < 0.01, all).

**Figure 4 nutrients-11-01517-f004:**
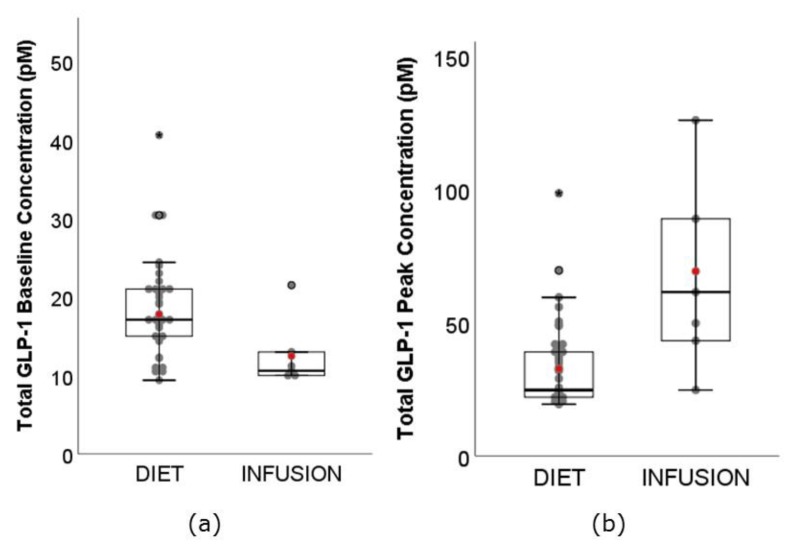
Boxplots showing the (**a**) baseline concentration, and (**b**) **C_max,_** peak concentration of total GLP-1 between DIET (N = 479, K = 37) and INFUSION (N = 67, K = 6). The weighted means were significantly different between DIET and INFUSION for all (*p* < 0.01, all).

**Figure 5 nutrients-11-01517-f005:**
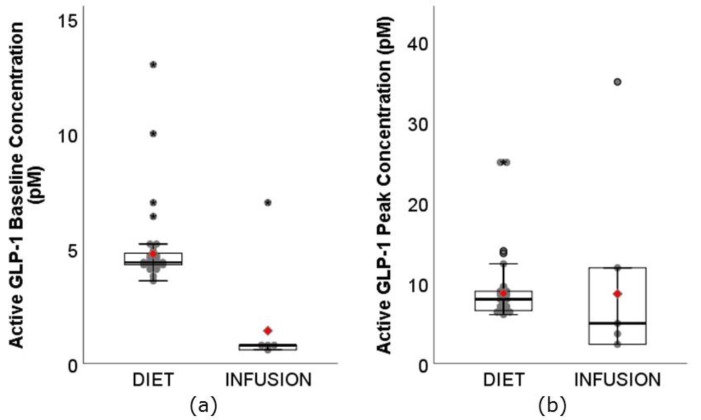
Boxplots showing the (**a**) baseline concentration, and (**b**) **C_max,_** peak concentration of active GLP-1 between DIET (N = 683, K = 29) and INFUSION (N = 81, K = 5). The weighted means were significantly different between DIET and INFUSION for (**a**) baseline concentration (*p* < 0.01), but not (**b**) **C_max,_** peak concentration (*p* = 0.96).

**Figure 6 nutrients-11-01517-f006:**
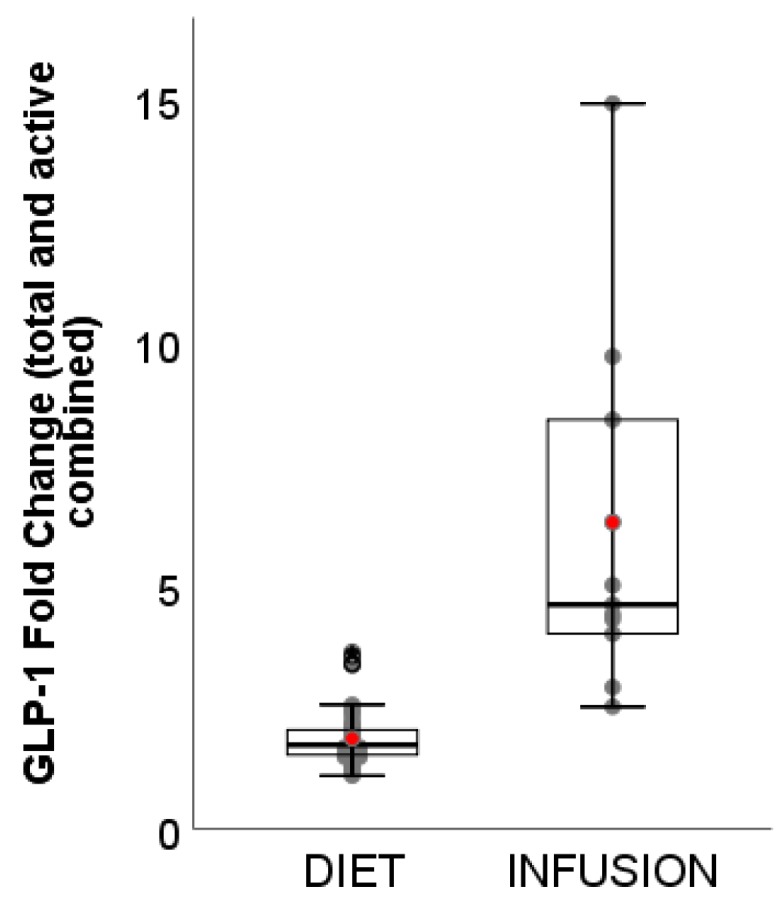
Boxplots showing fold change of GLP-1 (total and active) between DIET (N = 1162, K = 66) and INFUSION (N = 152, K = 11). The weighted means were significantly different between DIET and INFUSION (*p* < 0.01).

**Figure 7 nutrients-11-01517-f007:**
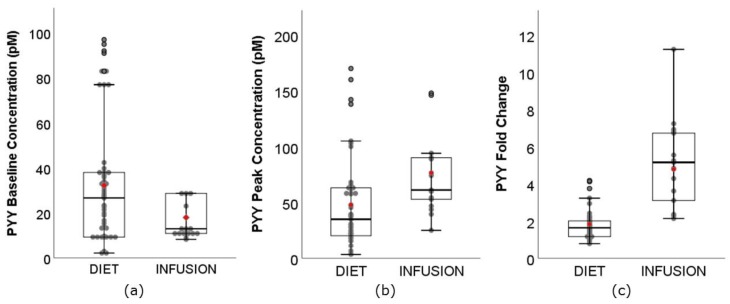
Boxplots showing (**a**) baseline concentration, (**b**) **C_max_**, peak concentration, and (**c**) fold change of PYY between DIET (N = 804, K = 50) and INFUSION (N = 132, K = 15). The weighted means were significantly different between DIET and INFUSION for all (*p* < 0.01, all).

**Table 1 nutrients-11-01517-t001:** Dietary preload (DIET) studies in lean men and women.

Reference	Gender	*N*	Study Duration (min)	Preload Meal	Format	Energy (MJ)	CHO (g)	Protein (g)	Fat (g)	Fold Change			Appetite Outcomes	
										CCK	GLP-1	PYY	Sensation	FI (kJ)
Nolan, et al. [23]	M	4	30	Tomato soup	NS	1.4	60	7	7	1.8	—	—	ND	—
	F	4	30	Tomato soup	NS	1.4	60	7	7	3.9	—	—	ND	—

Hall, et al. [24]	MF	9	90	Casein liquid meal	NS	1.7	24	48	11	2.8	1.7	—	FUL ^a^	3676 ^a^
	MF	9	90	Whey liquid meal	NS	1.7	20	40	9	3.2	2.0	—	FUL▲ ^b^	4537 ^b^

Bakhoj, et al. [25]	M	11	180	Einkorn honey salt bread	S	1.2	54	9	4	—	1.3	—	—	—
	M	11	180	Einkorn crushed WG bread	S	1.2	54	9	4	—	1.3	—	—	—
	M	11	180	Einkorn yeast bread	S	1.2	54	9	4	—	1.3	—	—	—
	M	11	180	Modern yeast bread	S	1.2	50	8	3	—	1.2	—	—	—

Frost, et al. [26]	MF	10	240	Control pasta	S	1.0	50	—	—	—	1.7	—	ND	4807
	MF	10	240	Fibre enriched pasta	S	1.0	50	—	—	—	1.4	—	ND	5167
	MF	10	240	Control pasta + fat	S	2.1	50	—	30	—	2.5	—	ND	4837
	MF	10	240	Fibre enriched pasta + fat	S	2.1	50	—	30	—	2.4	—	ND	4690

Pasman, et al. [27]	M	26	240	Simple CHO breakfast	S	1.8	80	12	7	2.5	—	—	STT ^a^	—
	M	26	240	Complex CHO breakfast	S	1.7	72	12	7	2.7	—	—	STT▲^b 2^	—

Adam [28]	MF	26	240	Glucose	NS	1.3	75	0	0	—	1.8	—	STT ^a^	ND
	MF	26	240	Glucose + guar gum	NS	1.3	75	0	0	—	1.5	—	STT▲ ^b^	ND

Sanggaard, et al. [29])	M	8	480	Whole milk	NS	3.7	62	48	49	5.0	3.4	1.9	ND	—
	M	8	480	Fermented milk + lactose	NS	4.1	81	52	49	3.7	3.6	2.3	ND	—

Burton-Freeman [30]	MF	25	45	Low fat shake	NS	1.1	52	10	1	2.0	—	—	STF ^a^	3014 ^a^
	MF	25	45	Safflower oil shake	NS	1.1	30	9	12	2.4	—	—	STF▲ ^b^	3198 ^a,b^
	MF	25	45	Walnut oil shake	NS	1.1	30	10	12	2.0	—	—	STF ^a^	3340 ^a,b^
	MF	25	45	Ground walnut shake	NS	1.1	30	9	12	2.3	—	—	STF ^a^	3457 ^b^

Adam and Westerterp-Plantenga [31]	MF	30	120	Breakfast	NS	1.9	55	31	12	—	1.7	—	ND	ND
	MF	30	120	Breakfast + galactose + guar gum	NS	2.7	105	31	12	—	3.6	—	ND	ND

Weickert, et al. [32]	F	14	300	Control bread	S	1.0	50	7	1	—	2.2	—	—	—
	F	14	300	Wheat fibre bread	S	1.0	50	7	1	—	1.7	—	—	—
	F	14	300	Oat fibre bread	S	1.0	50	7	1	—	1.9	—	—	—
Batterham, et al. [33]	MF	25	180	High protein pasta + dessert	S	4.6	47	178	21	—	—	2.0	HGR▼ ^a^	—
	MF	25	180	High CHO pasta + dessert	S	4.6	176	48	21	—	—	1.4	HGR ^b^	—
	MF	25	180	High fat pasta + dessert	S	4.6	46	46	80	—	—	1.8	HGR▼ ^a^	—

Blom, et al. [34]	M	15	180	High CHO plain yoghurt	NS	1.6	46	19	14	3.9	1.5	—	ND	5136
	M	15	180	High protein whey isolates	NS	1.7	14	57	12	7.9	2.0	—	ND	4697

le Roux, et al. [35]	MF	20	180	Liquid meal (500 mL)	NS	1.0	42	16	10	—	—	2.0	—	—
	MF	20	180	Liquid meal (500 mL)	NS	2.2	52	18	27	—	—	2.4	—	—
	MF	20	180	Liquid meal (500 mL)	NS	4.2	63	17	75	—	—	3.2	—	—
	MF	20	180	Liquid meal (900 mL)	NS	4.2	99	33	53	—	—	2.9	—	—
	MF	20	180	Liquid meal (900 mL)	NS	8.4	108	30	162	—	—	3.7	—	—
	MF	20	180	Liquid meal (900 mL)	NS	12.5	85	25	275	—	—	4.2	—	—

Weickert, et al. [36]	F	12	300	Control bread	S	1.0	50	7	1	—	—	1.5	ND	—
	F	12	300	Wheat fibre bread	S	1.0	50	7	1	—	—	1.1	ND	—
	F	12	300	Oat fibre bread	S	1.0	50	7	1	—	—	1.4	ND	—

Doucet, et al. [37]	F	25	180	Standard breakfast	S	2.4	82	19	19	—	—	1.7	—	2249

Smeets et al. [22]	MF	30	210	Adequate protein pasta	S	35% ER	60%	10%	30%	—	1.6	1.5	STT ^a^	—
	MF	30	210	High protein pasta	S	35% ER	45%	25%	30%	—	2.0	1.9	STT▲ ^b^	—

Sorensen, et al. [38]	M	20	285	Salatrim roll	S	3.2	97	19	40	2.4	1.7	1.3	FUL▲ ^a^	3414
	M	20	285	Margarine roll	S	3.2	97	19	40	2.4	2.1	1.6	FUL ^b^	3331

Zijlstra, et al. [39]^1^	M	12	90	Chocolate milk	NS	2.0	67	13	16	7.1	1.5	—	DTE ^a^, PCF ^a^	—
	F	20	90	Chocolate milk	NS	1.6	53	11	13					
	M	12	90	Chocolate custard	NS	1.9	64	13	17	5.3	1.5	—	DTE▼ ^b^, PCF▼ ^b^	—
	F	20	90	Chocolate custard	NS	1.5	50	10	13					

Hlebowicz, et al. [40]	MF	15	150	Rice pudding	NS	1.4	48	9	12	—	1.1	—	ND	—
	MF	15	150	Rice pudding + 1 g cinnamon	NS	1.4	48	9	12	—	1.4	—	ND	—
	MF	15	150	Rice pudding + 3 g cinnamon	NS	1.4	48	9	12	—	1.5	—	ND	—

Smeets and Westerterp-Plantenga [21]	MF	30	180	Lunch meal	S	35% ER	60%	10%	30%	—	1.6	2.2	ND	—
	MF	30	180	Lunch meal + red pepper	S	35% ER	60%	10%	30%	—	2.1	4.2	ND	—

Veldhorst, et al. [41]	MF	25	240	Whey custard	NS	2.5	82	15	23	—	2.0	—	STT▲ ^a^	2877
	MF	25	240	Whey custard	NS	2.5	82	37	13	—	2.1	—	STT ^b^	
	MF	25	240	Whey custard (No GMP)	NS	2.5	82	15	23	—	1.7	—	STT▲ ^a^	3208
	MF	25	240	Whey custard (No GMP)	NS	2.5	82	37	13	—	1.9	—	STT ^b^	

Veldhorst, et al. [42]	MF	25	240	Casein custard	NS	2.5	82	15	23	—	1.5	—	FUL ^a^	3133
	MF	25	240	Casein custard	NS	2.5	82	37	13	—	1.4	—	FUL▲ ^b^	3080

Veldhorst, et al. [43]	MF	25	240	Soy custard	NS	2.5	82	15	23	—	1.5	—	STT ^a^	3090
	MF	25	240	Soy custard	NS	2.5	82	37	13	—	1.5	—	STT▲ ^b^	3212

Nieuwenhuizen, et al. [44]	MF	24	240	α-lactalbumin custard	NS	2.5	82	15	23	—	2.0	—	ND	2650
	MF	24	240	Gelatine custard	NS	2.5	82	15	23	—	1.9	—	ND	2560
	MF	24	240	Gelatine custard + TRP	NS	2.5	82	15	23	—	2.0	—	ND	2610

Kohnke, et al. [45]	MF	11	360	Sandwich	S	2.3	34	12	40	2.1	—	—	—	—
	MF	11	360	Sandwich + 50 g thylakoid	S	3.0	45	35	45	2.0	—	—	—	—
	MF	11	360	Sandwich + 25 g thylakoid	S	2.6	40	24	42	1.3	—	—	—	—
	MF	11	360	Sandwich + 25 g delipidated thylakoid	S	2.6	39	24	41	1.4	—	—	—	—

Juvonen, et al. [46]	MF	20	180	Pudding	NS	1.3	57	4	4	—	—	1.1	ND	ND
	MF	20	180	Pudding with wheat bran	NS	1.3	55	6	4	—	—	1.1	ND	ND
	MF	20	180	Pudding with oat bran	NS	1.3	53	8	4	—	—	1.1	ND	ND
	MF	20	180	Pudding with wheat and oat bran	NS	1.3	57	7	4	—	—	1.1	ND	ND

Juvonen, et al. [47]	M	8	240	Viscous casein beverage	NS	1.0	4	54	0.5	—	1.9	—	ND	—
	M	8	240	Casein gel beverage	NS	1.0	4	54	0.5	—	1.4	—	ND	—
	M	8	240	Whey beverage	NS	1.0	4	52	0.3	—	3.6	—	ND	—

Brennan, et al. [48]	M	16	240	High fat pasta	S	3.8	68	34	55	1.2	—	1.8	—	4156 ^a,b^
	M	16	240	High protein pasta	S	3.8	68	101	25	1.4	—	1.6	—	3890 ^a^
	M	16	240	High CHO low protein pasta	S	3.8	135	23	30	1.6	—	1.7	—	4509 ^b^
	M	16	240	Adequate protein pasta	S	3.8	90	68	30	1.5	—	1.8	—	4533 ^b^

Kim, et al. [49]	F	10	180	Regular breakfast meal	S	2.1	77	26	11	—	—	0.8	—	—
	F	10	180	High protein breakfast meal	S	2.1	39	64	11	—	—	1.1	—	—
	F	10	180	High fat breakfast meal	S	2.1	39	26	28	—	—	0.9	—	—

Zhu, et al. [50]	M	19	180	Chicken soup with solid vegetable	NS	1.2	37	10	11	0.5	—	—	FUL ^a^, PWF ^a^	551.5 g
	M	19	180	Chicken soup with liquid vegetable	NS	1.2	37	10	11	0.8	—	—	FUL▲ ^b^, PWF▲ ^b^	545.6 g

van der Klaauw, et al. [51]	MF	8	270	High protein pancakes	S	1.7	20	60	9	—	1.5	1.4	ND	4280
	MF	8	270	High CHO pancakes	S	1.7	60	20	9	—	1.4	1.2	ND	4845
	MF	8	270	High fat pancakes	S	1.7	20	20	27	—	1.4	1.0	ND	4251

Little, et al. [52]	MF	16	180	Ensure Plus^®^ Nutrient Drinks	NS	3.1	100	32	25	2.8	—	1.6	—	3305

Zhu and Hollis [53]	M	8	180	Tomato basil soup	NS	1.6	62	4	13	1.1	—	—	—	—

Ohlsson, et al. [54]	MF	19	300	Breakfast + yoghurt (35 g oat oil)	NS	3.1	65	19	43	6.0	2.1	1.7	STT▲ ^a 3^	ND
	MF	19	300	Breakfast + yoghurt (0 g oat oil)	NS	2.9	65	19	39	5.3	1.8	1.7	STT ^b^	ND
	F	14	300	Breakfast + milk (14 g oat oil)	NS	2.9	65	19	39	6.1	3.5	2.2	—	ND
	F	14	300	Breakfast + milk (1.8 g oat oil)	NS	2.9	66	19	39	3.7	3.4	2.1	—	ND

Chungchunlam, et al. [55]	F	18	120	Maltodextrin preload drinks	NS	2.9	72	3	3	1.2	1.5	1.2	ND	2442 ^a^
	F	18	120	Whey preload drinks	NS	2.9	26	46	3	1.3	1.5	1.1	ND	2920 ^b^

Bligh, et al. [56]	M	19	180	Reference meal	S	1.6	57	13	11	—	1.8	1.0	HGR ^a^, FUL ^a^, DTE ^a^	—
	M	17	180	Palaeolithic meal 1	S	2.3	65	41	18	—	1.5	1.1	HGR▼ ^b^, FUL▲ ^b^, DTE▼ ^b^	—
	M	19	180	Palaeolithic meal 2	S	1.6	66	16	11	—	1.2	1.1	HGR▼ ^b^, FUL▲ ^b^, DTE▼ ^b^	—

Clamp, et al. [57]	M	10	180	Milkshake	NS	4.7	56	11	98	—	—	1.5	ND	533.7 g
	M	10	180	Milkshake	NS	4.7	56	11	98	—	—	1.4	ND	509.1 g

Hutchison, et al. [58]	M	20	180	70 g whey protein drink	NS	1.2	—	—	—	1.9	1.7	—	—	4176

Gonzalez-Anton, et al. [59]	MF	30	240	Cereal-based bread	S	1.1	38	13	4	—	1.7	—	PCF▼ ^a^, STT▲ ^a^	4184
	MF	30	240	White bread	S	1.0	47	7	4	—	1.9	—	PCF ^b^, STT ^b^	4268

Overduin, et al. [60]	MF	10	240	Sucrose control preload	NS	1.6	55	15	13	—	1.6	1.3	HGR ^a^	ND
	MF	10	240	Isovolumic erythritol preload	NS	1.2	29	14	29	—	2.3	1.7	HGR ^a^	ND
	MF	10	240	Isocaloric erythritol preload	NS	1.6	39	26	28	—	2.4	1.5	HGR▼ ^b^	ND

Appetite outcomes that are statistically different from each other are indicated by different superscript letters, i.e., a and b. Only subjective appetite outcomes that are statistically different when expressed in terms of Area under the Curve (AUC) are reported, unless otherwise specified. ^1^ Males and females received preload meal with different macronutrient composition but results for fold change were combined. ^2^ The difference in satiety was significant from T = 0 min to T = 90 min only. ^3^ The difference in satiety was significant for the female subgroup only. Abbreviations and symbols: M = Male, F = Female, MF = Mixed gender, S = Solid, NS = Non-solid, CHO = Carbohydrate, ER = Energy requirement, FI = Mean food intake reported in the unit of kJ except a few reported in the unit of g where specified, ND = No significant difference, FUL = Fullness, STT = Satiety, STF = Satisfaction, HGR = Hunger, DTE = Desire to eat, PCF = Prospective consumption of food, PWF = Preoccupation with food, ▲ = Significant increase, ▼ = Significant decrease, —= No data, not reported, no comparison or not assessed.

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
