# Peer review of "How Satiating Are the ‘Satiety’ Peptides: A Problem of Pharmacology versus Physiology in the Development of Novel Foods for Regulation of Food Intake"

_nutrients, 2019, doi:10.3390/nu11071517_

Round 1

Reviewer 1 Report

The authors have produced an excellent paper and I have just a few comments listed below which may further strength the paper.

Introduction - great, concise, well written. No change required.

Methods

- Search of only PubMed may have missed some studies; why were other relevant databases not searched e.g. EMBASE, web of science?

- List other specific inclusion criteria beyond those mentioned; e.g. english, oral preloads.

-  Other than baseline and peak conc, did you consider AUC or the conc immediately prior administration of the ad lib meal? Please justify why these were not considered.

- What was the rationale for excluding studies where study population was classified as overweight/obese, and also young adults? What age bracket defined young adults?

- Were there any other exclusion criteria used to reduce number of papers? e.g. non-english, not orally consumed, trained athletes.

- Remove double full stop at line 163

Results

- Was time to consumed preload/infusion recorded?

-Hutchison had a 30 g protein load as well; was it excluded for a reason?

- Not clear from table 1 what was the control intervention in each study listed; can information be added.

- In all of the box plot figures, stating what the dots on plots represent (outliers?) and what does K represent, may help other readers easily interpret.

Discussion

- Well constructed discussion that discusses all the major findings from the review very well.

- The data presented is from normal BMI, health adults so how well generalised are they to overweight/obese adults, and those with diseases like diabetes, cardiovascular disease.

- Can you add a comment on what a future study design should look like to better understand the relationship between satiety peptides and food intake. You mention they should measure at least 3 of satiety peptides. However, is it worth doing more acute trials like those reviewed, or should more effort go towards examine these relations over more chronic ad lib feeding trials?

Author Response

Reviewer 1

The authors have produced an excellent paper and I have just a few comments listed below which may further strength the paper.

Introduction - great, concise, well written. No change required.

Methods

- Search of only PubMed may have missed some studies; why were other relevant databases not searched e.g. EMBASE, web of science?

Author response: We agree that performing the search using the  PubMed database only may have resulted in fewer  studies. However, notably our current  review has increased the number of studies analysed considerably since the previous review conducted by Mars et al. (2012). With the updated data, the results still remain clear-cut and unambiguous. Although we do not think that the addition of further acute postprandial response studies is likely to change our main findings significantly, we acknowledge this limitation and this is now addressed in on page 19 Section 4.3,.

- List other specific inclusion criteria beyond those mentioned; e.g. english, oral preloads.

Author response: The inclusion of ‘oral preloads’ and ‘peptides infusion’ has already been addressed on Line 107. The selection of studies for English language only is now inserted on page 3 Line 108.

-  Other than baseline and peak conc, did you consider AUC or the conc immediately prior administration of the ad lib meal? Please justify why these were not considered.

Author response: Yes, we did consider AUC, but since AUC is affected by the total duration of measurement period, the longer measurement period (e.g. 240 mins) may have greater AUC than studies with shorter measurement period (e.g. 60 mins). Therefore, we concluded that AUC was not an appropriate measure to compare between studies, and hence we present baseline and Cmax change from baseline.

Although peptide concentration immediately prior to the administration of the ad lib meal is a sensible measure to predict the ad lib energy consumption, we did not record this concentration. Our main objective was to demonstrate that many (majority) preloads reported in the literature failed to achieve a peak peptide concentration that is comparable to infusion studies. Future review can potentially investigate the peptide concentration immediately prior to the administration of the ad lib meal. This is now being addressed on page 19 in Section 4.3.

- What was the rationale for excluding studies where study population was classified as overweight/obese, and also young adults? What age bracket defined young adults?

Author response: We included studies that were representative of an typical healthy and lean adult population only. Overweight/obese individuals have different peripheral circulating concentrations of GI peptides, though with some conflicting results (Steinert et al., 2017, MacLean et al., 2017, Zhu and Hollis, 2014). The exclusion of overweight/obese individuals served to remove any potential confounders to the observation. Similarly, studies that specifically focus on young adults or elderly could be a potential confounders to our observation(Giezenaar et al., 2018). We are not actively looking for a specific age range that defines young adults but we excluded based on the authors definition of young adults. The rationale is now briefly outlined in Line 121 and 122 on page 3, and is further addressed along with the generalisability of the review in Section 4.3, page 19.

- Were there any other exclusion criteria used to reduce number of papers? e.g. non-english, not orally consumed, trained athletes.

Author response: During our search, we did not actively include groups of individuals that did not represent a typical healthy and lean adult population, therefore ‘trained athletes’ was an exclusion criteria and now inserted on Line 120.

- Remove double full stop in Line 163

Author response: Double full stop is now removed.

Results

- Was time to consumed preload/infusion recorded?

Author response: Unfortunately, the time limit to consume the preload was not recorded in these publications. We believe that most preload studies were well-designed to provide a reasonable time period in order to  allow the participants to consume the meals freely to appetite, until comfortably full.

-Hutchison had a 30 g protein load as well; was it excluded for a reason?

Author response: It was excluded as a 30g protein preload contains less than 1MJ. This exclusion criteria has been outlined in Line 125 – 126, page 3.

- Not clear from table 1 what was the control intervention in each study listed; can information be added.

Author response: By definition, a control group should receive treatments identical to the experimental group with the exception of the investigational component of the meal. As such, there is often no true ‘control intervention’ in many preload studies. In fact, studies were designed to examine: (i) the dose-response relationship between macronutrients and peptide secretion (20% of studies); (ii) the differential effect of high carbohydrates, proteins or lipid on peptide secretion (15% of studies); (iii) the effect of different forms of protein, lipid or carbohydrates on peptide secretion (27.5% of studies); (iv) the effect of novel ingredients on peptides secretion (12.5% of studies); (v) miscellaneous studies which included comparisons between males and females, comparisons between lean and overweight, food format, and special diet such as commercial nutrient drinks and Palaeolithic diet (25% of studies).

- In all of the box plot figures, stating what the dots on plots represent (outliers?) and what does K represent, may help other readers easily interpret.

Author response: Each dot is representative of the mean of an intervention. This is now explained in Line 141, ‘K’ has been explained in Line 143, on page 4.

Discussion

- Well constructed discussion that discusses all the major findings from the review very well.

- The data presented is from normal BMI, health adults so how well generalised are they to overweight/obese adults, and those with diseases like diabetes, cardiovascular disease.

Author response: Generalisability of the review is now addressed in Section 4.3 on page 19, with particular focus around overweight/obese adults, individual variability, young vs old adults and men vs women. Unfortunately, the role of these peptides in diseased state is not well studied.

- Can you add a comment on what a future study design should look like to better understand the relationship between satiety peptides and food intake. You mention they should measure at least 3 of satiety peptides. However, is it worth doing more acute trials like those reviewed, or should more effort go towards examine these relations over more chronic ad lib feeding trials?

Author response: Our review showed that CCK, GLP-1 and PYY are poor predictors of appetite outcomes when being investigated independently. Although we do not have convincing evidence that measuring change in the  3 peptides concurrently will be able to better predict any change in appetite, it may still be useful to perform more acute trials to further explore the additive or synergistic mechanisms of peptides in the regulation of appetite. More chronic studies may also fill in the current research gap. This comment is now inserted in Section 4.3, page 19.

References:

GIEZENAAR, C., LUSCOMBE-MARSH, N. D., HUTCHISON, A. T., STANDFIELD, S., FEINLE-BISSET, C., et al. 2018. Dose-Dependent Effects of Randomized Intraduodenal Whey-Protein Loads on Glucose, Gut Hormone, and Amino Acid Concentrations in Healthy Older and Younger Men. Nutrients, 10.

MACLEAN, P. S., BLUNDELL, J. E., MENNELLA, J. A. & BATTERHAM, R. L. 2017. Biological control of appetite: A daunting complexity. Obesity (Silver Spring), 25, S8-S16.

MARS, M., STAFLEU, A. & DE GRAAF, C. 2012. Use of satiety peptides in assessing the satiating capacity of foods. Physiol Behav, 105, 483-8.

SADOUL, B. C., SCHURING, E. A. H., MELA, D. J. & PETERS, H. P. F. 2014. The relationship between appetite scores and subsequent energy intake: An analysis based on 23 randomized controlled studies. Appetite, 83, 153-159.

STEINERT, R. E., FEINLE-BISSET, C., ASARIAN, L., HOROWITZ, M., BEGLINGER, C., et al. 2017. Ghrelin, CCK, GLP-1, and PYY(3-36): Secretory Controls and Physiological Roles in Eating and Glycemia in Health, Obesity, and After RYGB. Physiol Rev, 97, 411-463.

ZHU, Y. & HOLLIS, J. H. 2014. Gastric emptying rate, glycemic and appetite response to a liquid meal in lean and overweight males. International Journal of Food Sciences and Nutrition, 65, 615-620.

Reviewer 2 Report

How satiating are the ‘satiety’ peptides: a problem of pharmacology versus physiology in the development of novel foods for regulation of food intake

The question posed by the authors of this study is interesting. The review is generally well written with good presentation of the data needed. I will proceed to a few suggestions to the authors.

Introduction

Introduction needs a clearer and more succinct format. Some information is reported more than once. I would suggest that after the first 2 paragraphs 1 paragraph on DIET studies and 1 paragraph on INFUSION is written. In this way the text will be much clearer.

Line 37: “…long been proposed” The phrase is shown more than 3 times in introduction. Also this sentence is much extended and has to be rewritten.

Line 51-54: The sentence is rather the outcome (except if it is cited) so I would suggest that it should be put at the end of introduction. Then again authors refer to the same in line 82 as well as in line 106-109.

Also in Figure 1 the subsequent Food Intake is missing as an outcome.

Methods

Methods are clear and detailed.

Line 160-163: This is the purpose of the study and should be at the end of introduction

Results

Tables are clear and detailed.

There are a few studies with missing data eg. Zhu &Hollis, 2014 appetite sensations not mentioned. Please check.

Discussion

Overall well written.

Although it is out of the scope of the review, it would good to refer to lean and obese phenotype responses (as shown in Clamp et al., 2015) and generally individuality/variability of responses in discussion.

Author Response

Reviewer 2

How satiating are the ‘satiety’ peptides: a problem of pharmacology versus physiology in the development of novel foods for regulation of food intake

The question posed by the authors of this study is interesting. The review is generally well written with good presentation of the data needed. I will proceed to a few suggestions to the authors.

Introduction

Introduction needs a clearer and more succinct format. Some information is reported more than once. I would suggest that after the first 2 paragraphs 1 paragraph on DIET studies and 1 paragraph on INFUSION is written. In this way the text will be much clearer.

Line 37: “…long been proposed” The phrase is shown more than 3 times in introduction. Also this sentence is much extended and has to be rewritten.

Line 51-54: The sentence is rather the outcome (except if it is cited) so I would suggest that it should be put at the end of introduction. Then again authors refer to the same in line 82 as well as in line 106-109.

Author response: We agree with the reviewer and we have revised the Introduction with the intent to decrease the redundancy. The motives of each paragraph in the Introduction are:

Paragraph 1 (Line 36 – 44) – To provide context about the value of developing novel food products for appetite control and that the understanding of physiological mechanisms in appetite regulation may help its success.

Paragraph 2 (Line 45 - 58) – To introduce the role of diet-induced secretion of GI peptides.

Paragraph 3 (Line 63 – 75) – To emphasise the importance of using an internationally accepted study design to produce clinical evidence to support health claims.

Paragraph 4 (Line 76 – 89) – To introduce the use of exogenous source of GI peptides to suppress appetite and the treatment of obesity and to forward the idea of pharmacological mechanism vs physiological mechanism.

Paragraph 5 (Line 90 – 99) – To inform about the conception of the study, aim and objectives.

Also in Figure 1 the subsequent Food Intake is missing as an outcome.

Author response: We agree that there is a good correlation between pre-meal subjective feelings of appetite (measured as mm VAS) and food intake (measured as energy intake), a minimum difference of 15mm in VAS is required to detect a significant difference in energy intake (Sadoul et al., 2014). However, we are being a little conservative by not stating food intake to be a direct outcome of subjective feelings of appetite. In our current review, we did not find that significant changes in subjective appetite outcomes were consistently associated with significant changes in food intake, especially in preload studies, although in-depth statistical analysis was not undertaken.

Methods

Methods are clear and detailed.

Line 160-163: This is the purpose of the study and should be at the end of introduction

Author response: We agree with the reviewer suggestion and these lines have been moved to the end of the Introduction (Line 96 – 100, page 3).

Results

Tables are clear and detailed.

There are a few studies with missing data eg. Zhu &Hollis, 2014 appetite sensations not mentioned. Please check.

Author response: Some studies have ‘missing’ data as these data were excluded as per the inclusion and exclusion criteria (Line 118 – 126, page 3). Zhu & Hollis (2014) compared the response of an identical preload meal between lean and obese individuals. The lean population is included in this review, but not the obese population. Therefore, appetite sensation data is ‘missing’ as there is no comparator to compare against in this review.

Discussion

Overall well written.

Although it is out of the scope of the review, it would good to refer to lean and obese phenotype responses (as shown in Clamp et al., 2015) and generally individuality/variability of responses in discussion.

Author response: We agree with the reviewer suggestion and this is now put forward in Section 4.3, page 19.
